# Different Patterns of Platinum Resistance in Ovarian Cancer Cells with Homologous Recombination Proficient and Deficient Background

**DOI:** 10.3390/ijms25053049

**Published:** 2024-03-06

**Authors:** Michela Chiappa, Federica Guffanti, Chiara Grasselli, Nicolò Panini, Alessandro Corbelli, Fabio Fiordaliso, Giovanna Damia

**Affiliations:** 1Department of Experimental Oncology, Istituto di Ricerche Farmacologiche Mario Negri IRCCS, Via M. Negri 2, 20156 Milan, Italy; michela.chiappa@marionegri.it (M.C.); federica.guffanti@marionegri.it (F.G.); chiara.grasselli@marionegri.it (C.G.); nicolo.panini@fht.org (N.P.); 2Department of Biochemistry and Molecular Pharmacology, Istituto di Ricerche Farmacologiche Mario Negri IRCCS, Via M. Negri 2, 20156 Milan, Italy; alessandro.corbelli@marionegri.it (A.C.); fabio.fiordaliso@marionegri.it (F.F.)

**Keywords:** ovarian cancer, drug resistance, cisplatin, isogenic cell system, DNA repair assays

## Abstract

Platinum compounds are very active in first-line treatments of ovarian carcinoma. In fact, high rates of complete remission are achieved, but most patients eventually relapse with resistant disease. Many mechanisms underlying the platinum-resistant phenotype have been reported. However, there are no data in the same isogenic cell system proficient and deficient in homologous recombination (HR) on platinum-acquired resistance that might unequivocally clarify the most important mechanism associated with resistance. We generated and characterized cisplatin (DDP)-resistant murine ovarian ID8 cell lines in a HR-deficient and -proficient background. Specific upregulation of the NER pathway in the HR-proficient and -resistant cells and partial restoration of HR in Brca1−/−-resistant cells were found. Combinations of different inhibitors of the DNA damage response pathways with cisplatin were strongly active in both resistant and parental cells. The data from the ID8 isogenic system are in line with current experimental and clinical evidence and strongly suggest that platinum resistance develops in different ways depending on the cell DNA repair status (i.e., HR-proficient or HR-deficient), and the upregulation and/or restoration of repair pathways are major determinants of DDP resistance.

## 1. Introduction

Ovarian cancer (OC) is one of the leading causes of gynecological cancer-related death, with more than 100,000 deaths/year in Western countries [1]. OC is classified in different subtypes, with high-grade serous carcinoma (HGSOC) being the most common and most lethal histotype [2]. Because of the relatively asymptomatic nature of OC and the lack of adequate screening tests, approximately 75% of patients are diagnosed with advanced disease, resulting in poor five-year overall survival. The standard treatment consists of cyto-reductive surgery followed by adjuvant platinum/taxol combined chemotherapy, with 70% of patients achieving complete remission after first-line therapy. Unfortunately, however, despite the initial regression, about half of the patients will relapse with resistant disease [3].

According to the Cancer Genome Atlas (TCGA), more than half of HGSOC have homologous recombination (HR) deficiency due to mutations in genes involved in this pathway, including *BRCA1* and *2* [4]. In 2005, two seminal studies demonstrated a synthetic lethality between poly-ADP-ribose polymerase inhibitors (PARPis) and HR deficiency [5,6], leading to FDA approval for the use of olaparib as maintenance therapy for platinum-sensitive *BRCA*-mutated HGSOC. PARP is are now approved in maintenance therapy after front-line platinum-based chemotherapy as well as in second- and third-line treatment of recurrent OC in HR-deficient or *BRCA*-mutated tumors [7].

Cisplatin (cis-diamminedichloroplatinum (II), DDP) is a metallic (platinum) coordination compound with square planar geometry, used as an anticancer drug. Cisplatin can covalently bind to DNA bases, forming inter- and intra-strand crosslinks [8]. The formation of those DNA adducts can block DNA synthesis and transcription, inducing DNA damage, cell cycle arrest, oxidative stress and finally apoptosis.

Different mechanisms underlying DDP resistance have been reported, including dysregulation of pumps/transporters such as the copper transporters 1 and 2 (CTR1 and CTR2), the ATPase copper-transporting alpha and beta (ATP7A and ATP7B) and the volume-regulated anion channels (VRAC) [9,10]. Platinum-induced DNA adducts are mainly repaired by nucleotide excision repair (NER), homologous recombination (HR) and mismatch repair (MMR), and the dysregulation of these pathways has been associated with DDP resistance [11]. Alterations in pathways that induce apoptosis, lower proteasomal degradation through protein de-ubiquitination and increase DDP-induced autophagy, intracellular drug detoxification and metabolic reprogramming have been reported in platinum-resistant cells [9,12]. About 80% of OC patients are very sensitive to platinum-based therapy, and only half of them bear HR deficiency, suggesting that HR is not the only determinant of DDP sensitivity and that the development of resistance could be different in a HR-proficient or HR-deficient background. To our knowledge, no data in isogenic cell systems proficient and deficient in HR are available on DDP-acquired resistance that could lead us to better understand the most important mechanisms associated with resistance. With this aim, we generated and characterized DDP-resistant murine HGSOC cell lines (ID8) in HR-deficient and -proficient backgrounds. ID8 cells were obtained from McNeish’s group [13,14], who engineered cells with CRISPR/Cas9 technology to delete only *TP53* (ID8 F3; HR-proficient), *TP53* and *Brca1* genes (ID8 Brca1−/−; HR-deficient). The main mechanisms of resistance were altered intracellular drug levels and the upregulation of the NER pathway in the HR-proficient F3-resistant cells and altered intracellular drug levels and the partial restoration of HR in Brca1−/−-resistant cells.

## 2. Results

### 2.1. Generation of DDP-Resistant Models

After exposing cells to increasing concentrations of cisplatin, we obtained the F3 ddpR subline from ID8 F3, >10 times more resistant (IC50 > 150 vs. 16.94 ± 8.26 µM, *p* = 0.003, Table 1 and Figure 1A), and the Brca1−/− ddpR subline from ID8 Brca1−/−, 5.7 times more resistant (IC50 22.05 ± 11.63 vs. 3.88 ± 0.86 µM, *p* < 0.02) than parental cells (Table 1 and Figure 1B). DDP resistance was confirmed via a colony formation assay (Figure 1C,D). While no differences in cell growth were observed between F3 and F3 ddpR sublines, the Brca1−/− ddpR cells showed slightly slower growth compared to the parental ID8 Brca1−/− (22.3 ± 1.93 vs. 20.0 ± 1.7 h, Appendix A). For both ddpR sublines, resistance was maintained for more than six months, in drug-free medium, with no changes in cell morphology detected in resistant cells compared to parental cells (Appendix A). All the subsequent experiments were conducted in this time frame.

### 2.2. Pharmacological Characterization

The pattern of cross-sensitivity/resistance to agents with different mechanisms of action can disclose underlying causes of resistance. For this reason, we pharmacologically characterized our resistant cell lines using agents with similar and different mechanisms of action from DDP and compared with them against parental ones. Both the ddpR sublines were cross-resistant to two other platinum-based compounds (carboplatin and oxaliplatin) (Table 1 and Appendix A). The Brca1−/− ddpR subline was cross-resistant to all the PARPi tested (olaparib, rucaparib and niraparib), while F3 ddpR cells were equally or more sensitive than parental cells (Table 1 and Appendix A). The fact that Brca1−/− ddpR cells were cross-resistant to PARPi suggests a possible restoration of HR.

The Brca1−/− ddpR subline was cross-resistant to Yondelis (ET-743), and a similar pattern, even if not reaching a statistical significance, was observed in F3 ddpR cells (Table 1 and Appendix A). Doxorubicin and paclitaxel had similar activity in Brca1−/− and Brca1−/− ddpR, while F3 ddpR cells were more sensitive to both drugs (Table 1 and Appendix A). 

The cytotoxic activities of different DNA damage response (DDR) inhibitors, KU55933 (ATM inhibitor), AZD6738 (ATR inhibitor), AZD7762 (Chk1 inhibitor), AZD1775 (Wee1 inhibitor) and DNA-PK inhibitor (NU7741), were also evaluated and reported (Table 1); in both the HR-proficient and -deficient backgrounds, DDP-resistant and parental cells gave a similar response to all five inhibitors, except for the DNA-PK inhibitor NU7741 (Table 1 and Appendix A) that was more active in Brca1−/− ddpR than in Brca1−/− cells. As treatments of DDR inhibitors could re-sensitize ovarian cells to cisplatin treatment independently from the HR status [15], we tested the combination of Wee1, Chk1 and ATR inhibitors with cisplatin in our models. All the combinations were synergistic in resistant and parental sublines, independent of the *Brca1* status (Appendix A). The combination of ATR and Wee1 was synergistic in all four cell lines (Appendix A).

### 2.3. DDP Induces Damage, Apoptosis and Different Pattern of Repair in Sensitive and Resistant Cells

We studied DNA damage induction, evaluating pSer139-H2AX, and apoptosis after the respective IC50 DDP doses in sensitive and resistant cells. In both F3 and Brca1−/− resistant sublines, p-H2AX induction was similar 24 and 48 h after DDP, compared to the parental cells (Figure 2A). A clear induction of the caspase 3/7 activity was observed in both parental cell lines, while a lower induction was observed in the ddpR cells (Figure 2B). 

Both HR and NER pathways have key roles in the repair of the DDP-induced DNA damage, and cells lacking these repair systems are extremely sensitive to the drug [11,16]. Secondary mutations in *BRCA1* and *2* genes have been described as one of the mechanisms of cisplatin resistance in *BRCA*-mutated cancers [17,18]. As RAD51 foci induction after DNA damage is a recognized functional test for HR proficiency [19], we recorded their induction after γ-irradiation in ID8 F3 and ID8 Brca1−/− parental and their DDP-resistant cells. While there was a slight increase in the number of RAD51 foci (not statistically significant) in F3 ddpR cells compared to parental F3 in basal conditions, RAD51 foci induction was comparable after IR (Figure 3A, Appendix A). In Brca1−/− cells, no RAD51 foci were detected at baseline (control untreated cells), or after damage; however, Brca1−/− ddpR cells presented a low but significant increase in the number of RAD51 foci vs. Brca1−/− cells already at the basal level (0 vs. 8.2, *p* = 0.017), with a further increase after IR treatment (Figure 3A). These data suggest similar (or slightly higher) HR activity in F3 ddpR cells than in F3 and the partial restoration of HR in Brca1−/− ddpR cells. Reversion mutation restoring BRCA1 activity can be excluded in Brca1−/− ddpR cells as they have a large *BRCA1* deletion (exon 12) [14].

As the reactivation of HR may be associated with an attenuation or decrease in the NHEJ repair pathway [20], we examined the mRNA levels of different genes involved in the upstream step of pathway choice (i.e., *REV7*, *53BP1* and *SHLD1*). The F3 ddpR subline showed downregulation of all three mRNAs (Figure 3B), while the Brca1−/− ddpR subline had a partial downregulation of *REV7* and *53BP1* but equal levels of *SHLD1* compared to the parental Brca1−/− cells (Figure 3C), partially supporting a preferential use of the HR pathway. When we examined the mRNA expression level of proteins involved in NHEJ, the minor differences in DNAPK, Ku70 and ligase IV levels observed between sensitive and resistance cells were not suggestive of an upregulation of NHEJ (Appendix A).

In order to assess the possible role of the NER pathway in DDP resistance, we first evaluated the UV sensitivity in parental and resistant sublines, as this type of damage is mainly repaired by NER [21]. F3 ddpR cells were much more resistant to UV exposure than their corresponding parental cells (Figure 3D), while Brca1−/− and Brca1−/− ddpR cells gave similar responses.

These data suggest an upregulation of the NER pathway in F3 ddpR cells, so to further corroborate this hypothesis, we indirectly measured the cellular NER activity via the host cell reactivation assay [22]. As detailed in Material and Methods, F3 parental and resistant cells were transfected with a mix of a UV-damaged luciferase reporter plasmid and a renilla plasmid, and after 24 h, both reporter expressions were recorded. After normalization, F3 ddpR cells had higher luciferase expression (Figure 3E). When we looked for the upregulation of proteins involved in the incision steps of the NER pathway, no change in ERCC1, XPF and XPG mRNA levels could, however, be detected between sensitive and DDP-resistant F3 cells (Appendix A).

These data suggest an upregulation of the NER pathway as a mechanism of DDP resistance in F3 ddpR cells and the partial restoration of HR in Brca1−/− ddpR cells.

### 2.4. Intracellular Drug Levels

As DDP resistance has been associated with decreased intracellular drug accumulation due to increased drug efflux, lower influx or both [9], we explored the role of different membrane pumps involved in DDP cellular transport. We investigated the transcript levels of the copper-transport efflux pumps ATP7A, ATP7B and CTR2 and the influx pump CTR1 whose upregulation and downregulation, respectively, have been associated with platinum resistance [23]. F3 ddpR cells showed a 3-fold downregulation of *ATP7A* and a 2.4-fold upregulation of *CTR2*, with no significant differences in *ATP7B* and *CTR1* mRNA levels (Figure 4A). In contrast, the expression levels of *ATP7A*, *ATP7B*, *CTR2* and *CTR1* were, respectively, three, four, seven and three times lower in Brca1−/− ddpR than in the corresponding parental cells (Figure 4B). mRNA levels of the *MDR-1* gene, which codes for the efflux pump responsible for the excretion of different anticancer drugs and whose increased levels have been reported in patients with platinum-resistant ovarian tumors, were lower in the F3 ddpR cells than parental ID8 F3 cells (Figure 4A), while there was a 3.4-fold upregulation in the Brca1−/− ddpR subline compared to the parental cells (Figure 4B). Besides transport pumps, DDP may enter the cell through ionic channels [10], so we evaluated the mRNA level of the volume-regulated anion channel *LRRC8D* gene: downregulation was clear in both F3 and Brca1−/− ddpR sublines compared to the corresponding parental cells (Figure 4A,B). Taken together, these data suggest slightly reduced DDP intracellular levels in both resistant sublines.

### 2.5. Metabolic Stress

As the acquisition of DDP resistance has been associated with metabolic reprogramming [24], we explored the metabolic asset of our parental and resistant cells. First, we looked at the levels of intracellular ROS in our models at the basal level and found no differences between F3 and Brca1−/− parental cells and between F3 parental and its corresponding resistant cells but slightly higher levels of ROS in Brca1−/− ddpR cells compared to Brca1−/− (Figure 5A and Appendix A). Considering that mitochondria produce ROS [25], we then tested the effect of metformin, a commonly used oral anti-hyperglycemic agent for type II diabetes with inhibitory action on the mitochondrial electron transport chain complex I [26]. ID8 Brca1−/− ddpR cells were ten times more sensitive to metformin than the parental ID8 Brca1−/− (IC50 2.87 ± 1.04 vs. 22.15 ± 6.96 mM, *p* = 0.009) (Figure 5B). In line with this, Brca1−/− ddpR cells were also 29 times more sensitive to the metformin analog fenformin (IC50 0.11 ± 0.07 vs. 3.14 ± 1.65 mM, *p* = 0.04) and 100 times to the specific complex I inhibitor rotenone (IC50 3.88 ± 1.39 vs. 388.5 ± 16.26 nM, *p* = 0.0009) (Appendix A). No difference in metformin and rotenone cytotoxicity could be observed between F3 and its corresponding resistant cells, while F3 ddpR cells were 3.5 times more sensitive to fenformin (Appendix A). A further ROS increase, even if not statistically significant, was observed only in Brca1−/−ddpR cells after the IC50 dose of metformin (Figure 5A) and not in F3 ddpR (Appendix A). Given these findings, we explored mitochondrial involvement via electron microscopy. No mitochondrial alterations could be observed in F3 and F3 ddpR cells (Appendix A); on the contrary, we found significant mitochondrial ultrastructural alterations characterized by peripheral matrix condensation, cristolysis and massive swelling in Brca1−/− ddpR compared to Brca1−/− cells (53.5% ± 8.0% vs. 3.4% ± 1.1%, *p* < 0.0001, Figure 5C). Mitochondrial impairment in Brca1−/− ddpR cells was functionally confirmed by using Seahorse technology, where a lower ATP production mainly at the mitochondrial level could be found in Brca1−/− ddpR cells as compared to the parental cells (Figure 5D,E and Appendix A). These ultrastructural and functional mitochondrial impairments are not associated with an increased basal level of apoptosis in Brca1−/− ddpR compared to Brca1−/− (Appendix A).

### 2.6. In Vivo Activity

As detailed in Materials and Methods, both resistant and parental sublines were transplanted in immunocompetent C57Bl/6 mice. No differences in tumor take were observed. F3 ddpR-transplanted mice had longer survival (log-rank test, *p* < 0.0084) than those with F3 cells, and the opposite was seen for Brca1−/−-sensitive and -resistant transplanted mice (log-rank test, *p* < 0.003). DDP was slightly active in parental F3 cells (ILS = 39.6%, log-rank test, *p* < 0.003) and completely inactive in the F3 ddpR subline (ILS = −13.6%, log-rank test) (Figure 6A). Again, DDP was very active in mice transplanted with Brca1−/− cells (ILS = 72%) and less active in the Brca1−/− ddpR subline (ILS = 45%) (Figure 6B). These data suggest that cells made resistant in vitro to DDP retain drug resistance in vivo.

## 3. Discussion

DDP is one of the most widely used chemotherapeutic agents for the treatment of different solid tumors, including ovarian cancer. For ovarian cancer, platinum compounds are the therapeutic gold standard in combination with taxol, with 70% of patients achieving complete remission after this first-line therapy [27]. This striking activity has been associated with a defect in HR-mediated DNA repair [28] seen in more than half of the HGSOCs. Besides the initial high response rate, however, most of patients eventually relapse with platinum-resistant disease [3]. There is, therefore, an urgent need to clarify the basis of acquired platinum resistance so as to develop therapeutic options to overcome/delay this resistance. Considering that HR deficiency is a hallmark of HGSOC and that HR is involved in the repair of DDP-induced DNA lesions [11], we generated cisplatin-resistant murine ovarian cells from HR-proficient and -deficient backgrounds through multiple in vitro treatments with increasing drug doses, using ID8 cells engineered with CRISPR/Cas9 technology to delete the *TP53* gene (F3) or both *TP53* and *Brca1* genes (Brca1−/−) [13,14]. Our results can be summarized as follows: (i) we obtained stable DDP-resistant sublines both from HR-proficient and -deficient backgrounds; (ii) the main mechanisms of resistance were altered intracellular drug levels and upregulation of the NER pathway in the HR-proficient F3-resistant cells and altered intracellular drug levels and partial restoration of HR in Brca1−/−-resistant cells; (iii) Brca1−/− ddpR cells had higher ROS content, presented structural mitochondrial alterations and were very sensitive to metformin; and (iv) resistant sublines maintained DDP resistance also when transplanted in vivo.

The F3 and Brca1−/− ddpR sublines were, respectively, >10 and 5.7 times more resistant to DDP than the corresponding parental cells, and this resistance was maintained for more than six months with no further drug treatment. Both ddpR sublines were cross-resistant to carboplatin and, to a lesser degree, to oxaliplatin, supporting published data [29].

DNA repair capacity is one of the most important determinants of cancer cell sensitivity/resistance to DNA-damaging agents, including DDP [30,31]. Platinum-induced DNA damage is mainly repaired by the NER, HR and Fanconi anemia pathways [11]. The pharmacological pattern of cross-resistance and the molecular characterization of our resistant models revealed a preferential upregulation of the NER pathway in the HR-proficient background and restoration of the HR pathway in the HR-deficient DDP-resistant cells. Regarding F3 ddpR cells, both the relative resistance to UV damage and the higher expression of a UV-induced damage luciferase reporter plasmid (read out of a functional NER [22]) compared to parental F3 cells strongly suggest an upregulated NER activity that could partially explain both the DDP resistance and the platinum analogs’ cross-resistance. NER alterations have been reported in 8% of HGSOC with similar overall survival and progression-free survival as in *BRCA1/2*-mutated patients, supporting the fact that NER inactivation confers enhanced DDP sensitivity [32] as many in vitro findings suggest [33].

While the inactivation of NER has been clearly associated with extreme DDP sensitivity, it has been more difficult to associate its upregulation with DDP resistance, mainly for the lack of NER activity functional tests. There are many contrasting data on ERCC1, a key protein in the NER pathway, as a predictive biomarker for a response to platinum-based therapy in different tumors, including OC [34,35]. Few data, however, are available on the use of the host cell reactivation assay that can be considered a proxy of NER activity. Our data strongly suggest that F3 ddpR has a functional increase in NER activity.

Interestingly, only Brca1−/− ddpR was cross-resistant to PARPi, suggesting a possible restoration of HR, as reported in [36], that was demonstrated by the increase in RAD51 foci formation after DNA damage. The restoration of HR in BRCA-mutated tumors due to reversion mutation has already been described as a mechanism of DDP resistance in cell lines and clinical samples [17,18]. This can also be proposed for PARPi [36]. In our experimental conditions, reversion mutations were very unlikely as these cell lines were generated by CRISPR/Cas9 technology with the deletion of the entire exon 2, preventing the acquisition of reversion mutations to restore the frame of an active truncated BRCA1 protein. It has been shown that HR can be restored by the downregulation of proteins favoring NHEJ repair and funneling DNA double-strand repair to the HR repair pathway. This is what seemed to happen in our Brca1−/− ddpR cells, as indirectly suggested by the downregulation of *REV7* and *53BP1*. Lower levels of these proteins have in fact been associated with an increase in the end resections at the double-strand break, favoring/promoting HR [37,38,39].

The DNA damage response (DDR) machinery has a fundamental role in maintaining genomic stability and, in some cases, presents a cancer-specific vulnerability exploited by synthetic lethality-based therapies [40]. As expected, the DDR inhibitors showed greater cytotoxic activity in Brca1−/− cells than in F3 cells; however, limited cross-resistance was found only in F3 ddpR cells. When these inhibitors were combined with cisplatin, these co-treatments were similarly synergic in all four cell lines. These data suggest that the inhibition of ATM/ATR/Chk1/Wee1 may offer a therapeutic strategy in OC independently from the HR-proficient/deficient background both as single agents and in combination with a platinum drug. Phase I studies of the combination of platinum compounds and inhibitors of ATR [41,42,43], Wee1 [44] and Chk1 [45,46] in solid tumors are already available, and further clinical investigations are warranted with schedule refinement to help manage toxicity. Interestingly, the combination of ATR/Wee1 inhibitors was also synergic in both sensitive and resistant F3 and Brca1−/− cells.

Low intracellular levels of DDP have been described as a possible mechanism of drug resistance [9,23]. MDR1 is a cell surface efflux pump involved in the excretion of different anticancer drugs, and even though DDP is not its direct substrate, its upregulation has been reported in samples from patients with DDP-resistant OC [9]. The *MDR1* gene was upregulated 3.4 times in the Brca1−/− ddpR subline, though there was no cross-resistance to paclitaxel or doxorubicin, and two direct substrates of MDR1 were observed, quite likely because much higher (100 fold) MDR1 levels are necessary [47]. Copper transporters have been reported to have a role in DDP intracellular levels and consequently in cisplatin cytotoxicity [12]. Blair et al. described a correlation between higher mRNA levels of CTR2 efflux pumps and lower sensitivity to DDP in a panel of OC cell lines [48]. We too observed a 2.4-times upregulation of *CTR2* in the F3 ddpR cells that may explain the lower DDP intracellular levels and the lower cytotoxic effect. Downregulation of both *CTR1* and *LRRC8D* expression has been associated with lower DDP intracellular levels and cytotoxicity [10,49]. We found a 3-fold *CTR1* decrease in Brca1−/− ddpR cells and 2.3- and 6-fold *LRCC8D* decreases, respectively, in F3 and Brca1−/− ddpR cells, suggesting a lower influx of cisplatin.

Brca1−/− ddpR cells showed high intracellular ROS levels and high sensitivity to metformin, fenformin and rotenone, suggesting metabolic alteration. As we previously demonstrated mitochondrial structural alterations in OC patient-derived xenografts made resistant to DDP [50], we employed electronic microscopy on our in vitro model, confirming the presence of damaged mitochondria in Brca1−/− ddpR cells. Seahorse analyses confirmed an impairment of mitochondrial activity and less ATP production. It was not clear how these cells could compensate energetically, and an increase in fatty acid production [51] could be an alternative aspect to study. Several studies have proposed targeting mitochondrial metabolism as a therapeutic strategy in cancers cells [24], including a Phase II clinical trial of metformin as an anti-cancer agent in OC [52]. Our data foster the development of this approach in DDP-resistant tumors. Finally, when we transplanted our DDP-sensitive and -resistant cell lines in immunocompetent C57bl/6 mice, both F3 and Brca1−/− ddpR sublines were resistant to DDP, also in vivo.

We are aware of the fact that our resistant cell lines were obtained in vitro using DDP concentrations and a time of exposure that might not perfectly mimic the clinical setting. However, we think that the data we obtained are of value as, using different DDP doses (high in HR-proficient and low in HR-deficient cells), we were able to disclose different mechanisms of resistance in the two backgrounds, and in both cases, different mechanisms developed, supporting the enormous plasticity of tumor cells undergoing treatment.

## 4. Materials and Methods

### 4.1. Cisplatin-Resistant Clones

ID8 F3 (*Tp53−/−*) and ID8 Brca1−/− (*Tp53−/−*; *Brca1−/−*), indicated here, respectively as F3 and Brca1−/−, were kindly provided by I.A. McNeish (Institute of Cancer Sciences, University of Glasgow, Glasgow, UK) [13,14] and maintained in DMEM medium (Gibco, Life Technologies, Carlsbad, CA, USA) supplemented with 2 mM glutamine, 4% FBS, 5 µg/mL insulin, 5 µg/mL transferrin and 5 ng/mL sodium selenite, at 37 °C with 5% CO_2_. Cisplatin resistance was induced by treating cells with increasing drug concentration (from 10 to 20 µM for ID8 F3 and from 2.5 µM to 10 µM for ID8 Brca1−/−). ID8 F3 cisplatin-resistant (F3 ddpR) were obtained in a time frame of six months, while ID8 Brca1−/− cisplatin-resistant (Brca1−/− ddpR) required one year to be established.

### 4.2. Cell Growth

Growth curves were obtained by seeding the cells at 5000 cells/mL in six-well plates and counting them with the Multisixer 3 Coulter Counter (Beckman Coulter, Brea, CA, USA) at different times. For colony assays, cells were seeded at 150 cells/mL in six-well plates; after 48 h, the cells were treated with increasing doses of cisplatin. After 10 days, the colonies were stained with Gram’s crystal violet solution (Merck, Darmstadt, Germany).

### 4.3. Drugs and Treatments

Cisplatin, metformin and fenformin were purchased from Sigma-Aldrich (St. Louis, MO, USA); carboplatin from Adipogen (San Diego, CA, USA); paclitaxel from ChemieTek (Indianapolis, IN, USA); doxorubicin and rotenone from Merck; Yondelis (ET-743) from PharmaMar (Madrid, Spain); olaparib from TargetMol (Boston, MA, USA); oxaliplatin, rucaparib, niraparib, KU55933 (ATM inhibitor), AZD6738 (ATR inhibitor), AZD7762 (Chk1 inhibitor) and AZD1775 (Wee1 inhibitor) from Axon Medchem (Groningen, The Netherlands). All the drugs were dissolved in DMSO or water as stock solutions and diluted in medium just before treatment. For cytotoxicity experiments, cells were seeded at 1000–2000 cells/mL and treated with different drug concentrations in 96-well plates 48 h after seeding. After five days of treatment, cell viability was examined with the MTS assay system (Promega Corporation, Madison, WI, USA), and absorbance was acquired using a microplate reader (GloMax Discover, Promega Corporation). Drug concentrations inhibiting growth in 50% of the cells (IC50) were calculated for each cell line, with the interpolation method on Prism 9.5.1 (GraphPad Software, La Jolla, CA, USA). All the experiments were run at least three times in sestuplicate.

### 4.4. Western Blot

Cell pellets were lysed for 30 min in ice-cold extract buffer (50 mM TrisHCl), pH 7.4, 250 mM NaCl, 0.1% Nonidet NP40, 5 mM EDTA, 50 mM NaF and a protease inhibitor cocktail (Sigma-Aldrich). Lysates were cleared by centrifuging at 12,000 rpm for 15 min, and the protein concentration was determined using a Bio-Rad assay kit (Bio-Rad Laboratories S.r.l., Hercules, CA, USA). Cell lysates (50 µg) were resolved on 10–12% SDS-PAGE (polyacrylamide gel electrophoresis) gels. Proteins were then transferred to nitrocellulose membranes (Merck Millipore, Burlington, MA, USA). Immunoblotting was conducted with the following antibodies: rabbit anti-γH2AX Cat#9718 (Cell Signaling, Danver, MA, USA, 1:1000) and mouse anti-ACTIN sc-47778 (Santa Cruz Biotechnology, Dallas, TX, USA, 1:500). The secondary antibodies conjugated with horseradish peroxidase (HRP) goat anti-rabbit Cat#170-6515 and goat anti-mouse Cat#170-6516 were purchased from Bio-Rad Laboratories S.r.l. The HRP substrate (ECL Western Blotting Detection, Amersham-Life Science, Amersham, UK) was added, and the signal was detected with the Odyssey Fc instrument (Li-COR, Lincoln, NE, USA). The uncropped version of the gels are shown in Appendix A.

### 4.5. Caspase 3/7 Activity

Caspase-3/7 activity was measured with the Caspase-Glo^®^3/7 kit (Promega Corporation). Briefly, cells were seeded at 10,000 cells/mL in white flat-bottom 96-well plates and, after 24 h, were treated with DDP 24, 48 and 72 h; after this, Caspase Glo reagent was added to the culture medium and incubated at 37 °C for 45 min. Then, luminescence was read using a microplate reader (GloMax Discover, Promega Corporation). Caspase activity was expressed as mean relative light units (RLUs) and normalized over the cell viability.

### 4.6. Quantitative Reverse Transcription (RT)-PCR

Total RNA from cells was purified with the Maxwell 16 LEV SimplyRNA (Promega Corporation) and retro-transcribed with the High-Capacity cDNA Reverse Transcription Kit (Applied Biosystems, Waltham, MA, USA). Gene expression was measured by quantitative real-time PCR with SYBR green technology (Applied Biosystems) using ad hoc-designed primers (Appendix A). The real-time PCRs were run in triplicate. All data were normalized to the levels of the β-actin gene and analyzed using the ΔΔCt method.

### 4.7. RAD51 Immunofluorescence

Cells were seeded on coverslips in 24-well plates at 15,000 cells/mL, irradiated (10 Gy) after 24 h and then fixed in 5% paraformaldehyde for 30 min. Cells were permeabilized with 0.2% Triton (Sigma-Aldrich) in PBS for 15 min and stained with rabbit anti-RAD51 ab63801 (Abcam, Cambridge, UK) diluted 1:1000 in blocking solution (BSA 5%). Nuclei were stained with 4′,6-diamidino-2-phenylindole (DAPI) (30 ng/mL in PBS, Sigma-Aldrich). Slides were mounted with Vectashield solution (VectorLab, Newark, CA, USA) and observed using the Nikon Instruments A1 Confocal Laser Microscope, with the Plan Fluor 40× DIC M N2 NA = 0.8 WD = 660 μM objective. RAD51 foci were quantified by scoring cells with five or more foci per nucleus. At least five areas in the z-stacking of each sample were acquired and analyzed with ImageJ FIJI 2.1.0/1.53t software (Bethesda, MD, USA).

### 4.8. Host Cell Reactivation Assay

pGL4.53[luc2/TK] reporter plasmid (Promega Corporation) was diluted in TE buffer and pipetted onto different 60 mm culture dishes to be irradiated with UVB at 0, 10, 25, 50 and 100 J/m^2^. After UV treatment, plasmids were precipitated and dosed using a NanoDrop spectrophotometer (Thermo Fisher Scientific, Waltham, MA, USA). Untreated pRL-TK plasmid (Promega Corporation) was used for normalization. F3 and F3 ddpR cells were seeded at 10,000 cells/mL in white flat-bottom 96-well plates and transfected after 24 h with 800 ng/well of plasmid mixes containing the UV-treated-pGL4.53 and pRL in a ratio of 1:10 using the Lipofectamin 2000 (Invitrogen, Carlsbad, CA, USA). After 24 h of transfection, the luciferase/renilla expressions were evaluated using the Dual-Glo^®^ Luciferase Assay System (Promega Corporation). Data are expressed as % of luminescence over untreated pGL4.53 reporter plasmid and are the mean of two different experiments run in four replicates.

### 4.9. ROS Detection

For ROS analyses, 500,000 cells were stained with the OxiSelect In Vitro ROS/RNS Assay Kit (Cell Biolabs, Inc., San Diego, CA, USA). Briefly, two days before the assay, 20,000 cells/mL were seeded in 6-well plates and, after 24 h, treated with metformin. After another 24 h, cells were permeabilized with 0.5% Triton X-100 (Sigma-Aldrich) in PBS for 30 min and seeded in black flat-bottom 96-well plates in triplicate and stained with DCFH solution. Cell fluorescence was read using a microplate reader (GloMax Discover, Promega Corporation).

### 4.10. Seahorse Analyses

For seahorse analyses, ID8 Brca1−/− and Brca1−/− ddpR cells were seeded in XF-24 plates at a density of 10,000 cells/well. After 24 h, the culture medium was replaced with 500 μL of XF DMEM medium (Agilent Seahorse, Santa Clara, CA, USA), and the plates were incubated at 37 °C in a non-CO2 incubator for 1 h. For ATP production analyses, we used the Seahorse XF Real-Time ATP Rate Assay Kit (Agilent Seahorse) and the XFe24 Analyzers (Agilent Seahorse). At the end of the assay, cells were fixed in 5% paraformaldehyde for 30 min and stained with 4′,6-diamidino-2-phenylindole (DAPI, Sigma-Aldrich). Three areas of each well were acquired and quantified using ImageJ win-64 software for data normalization.

### 4.11. Transmission Electron Microscopy

ID8 Brca1−/− and Brca1−/− ddpR cells were centrifuged, washed with phosphate buffer (0.12 M) and fixed overnight with 4% paraformaldehyde and 2% glutaraldehyde. Cell pellets were then dissected in 0.5–1 mm 3 blocks and fixed with 1% (wt/vol.) OsO4 in cacodylate buffer (0.12 M) for 2 h at RT. After dehydration in a graded series of ethanol preparations, pellets were cleared in propylene oxide, embedded in epoxy medium (Epoxy Embedding Medium Kit; Sigma-Aldrich) and polymerized at 60 °C for 72 h. From each sample, a 1 μm section was cut with a Leica EM UC6 ultramicrotome (Leica Microsystems, Wetzlar, Germany), stained with toluidine blue and mounted on glass slides. Ultra-thin 60 nm sections were obtained and counterstained with uranyl acetate and lead citrate, and images were obtained via an energy filter transmission electron microscope (Libra120, Carl Zeiss, Oberkochen, Germany) coupled with a yttrium aluminum garnet (YAG) scintillator slow-scan CCD camera (Sharp eye, TRS). Mitochondria were considered damaged if they presented at least one of either peripheral matrix condensation, cristolysis or massive swelling. 

To calculate the percentage of damaged mitochondria, we analyzed more than 350 mitochondria in at least 25 different cells for each experimental group on images acquired via the iTem software (version 5.0 Build 1223, Olympus Soft Imaging Solutions, Germany).

### 4.12. Flow Cytometry

For the apoptosis assay, 500,000 cells were stained with the FITC Annexin V/PI Apoptosis Detection Kit (BioLegend, San Diego, CA, USA), according to the manufacture’s protocol. Cellular fluorescence was quantified via flow cytometry (Cytoflex LX, Beckman Coulter, Brea, CA, USA), and offline analysis was performed using Kaluza software version 2.1 (Beckman Coulter).

### 4.13. In Vivo Studies

Furthermore, 6 × 10^6^ cells were injected intraperitoneally in C57Bl/6 female mice (Charles Rivers Laboratories, Wilmington, MA, USA). After 7 days, mice were randomized to receive DDP (5 mg/kg q7 × 3, i.v.) or vehicle. Body weights were recorded twice a week, and tumor growth was assessed by evaluating abdominal distension. Antitumor activity was evaluated by calculating the increase in lifespan (ILS%) = [(median survival days of treated mice − median survival days of control mice)/median survival days of treated mice] × 100. The IRFMN adheres to the ethical principles set out in the following laws, regulations, and policies governing the care and use of laboratory animals: Italian Governing Law (D.lgs 26/2014; Authorization n.19/2008-A, issued 6 March 2008, by the Ministry of Health); Mario Negri Institutional Regulations and Policies providing internal authorization for persons conducting animal experiments (Quality Management System Certificate—UNI EN ISO 9001:2015—Reg. 6121, https://www.iso.org/standard/62085.html (accessed on 23 February 2024)); the NIH Guide for the Care and Use of Laboratory Animals (2011 edition); and EU directives and guidelines (EEC Council Directive 2010/63/UE). An institutional review board and the Italian Ministry of Health approved the present project (approval 87/2018-PR).

### 4.14. Statistical Analysis

Statistical analysis was conducted with GraphPad Prism 9.5.1 (GraphPad Software), using the tests specified in the legends to the figures.

## 5. Conclusions

In summary, we generated two DDP-resistant ovarian cancer cell lines in HR-proficient and -deficient backgrounds with different mechanisms of resistance. F3 ddpR cells showed upregulation of the NER pathway, and Brca1−/− ddpR cells showed a partial restoration of the HR pathway. In both resistant sublines, there were alterations of cisplatin efflux–influx pumps, suggesting low intracellular drug levels. The resistance was maintained in vivo. The data from the ID8 isogenic system are in line with the experimental evidence and strongly suggest that platinum resistance develops in different ways depending of the cell DNA repair status (HR-proficient or HR-deficient) and that the upregulation and/or restoration of repair pathways are a major determinant of DDP resistance.

## Figures and Tables

**Figure 1 ijms-25-03049-f001:**
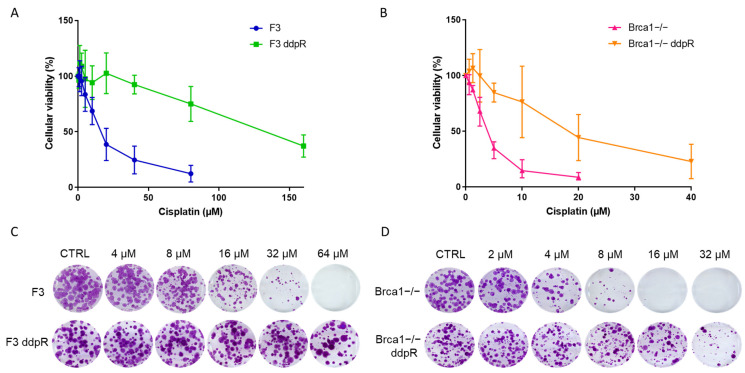
In vitro characterization of F3, F3 ddpR, Brca1−/− and Brca1−/− ddpR cells. Dose–response curves of cisplatin in F3 (blue line) and F3 ddpR (green line) cells (**A**) and Brca1−/− (pink line) and Brca1−/− ddpR (orange line) cells (**B**). Data are the mean ± standard deviation (SD) of three independent experiments run in sestuplicate. (**C**) Colony assay of ID8 F3 and F3 ddpR untreated and after DDP 4 µM, 8 µM, 16 µM, 32 µM and 64 µM. (**D**) Colony assay of ID8 Brca1−/− and Brca1−/− ddpR untreated and after DDP 2 µM, 4 µM, 8 µM, 16 µM and 32 µM. Magnification 1×.

**Figure 2 ijms-25-03049-f002:**
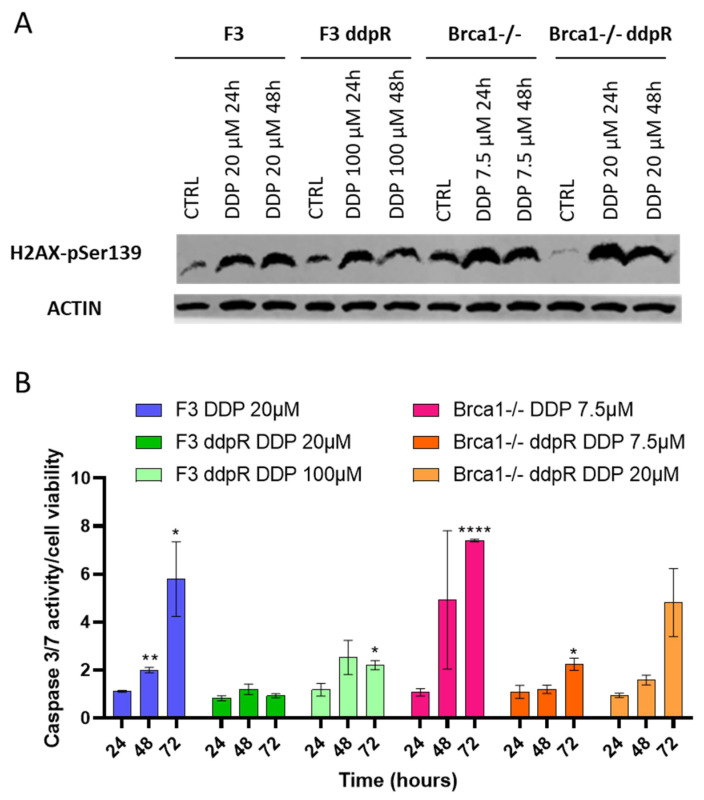
DNA damage and apoptosis. (**A**) Western blot analysis of pSer139-H2AX 24 and 48 h after DDP treatment in F3 and in Brca1−/− parental and resistant sublines. (**B**) Caspase 3/7 activity in F3 (blue), F3 ddpR (green), Brca1−/− (pink) and Brca1−/− ddpR (orange) cells 24, 48 and 72 h after DDP. Data are expressed as the fold increase over untreated cells and are the mean ± SD of two independent experiments. For statistical analyses, unpaired *t*-test was used. Only statistically significant differences are reported: * = *p* < 0.05; ** = *p* < 0.005; **** = *p* < 0.0001.

**Figure 3 ijms-25-03049-f003:**
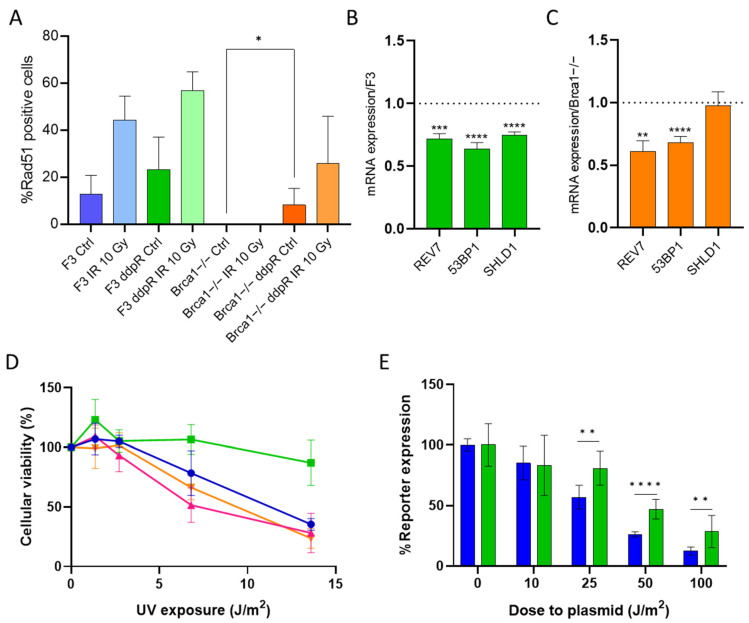
DNA repair in cisplatin-sensitive and -resistant cells. (**A**) Percentage of Rad51 positive cells treated or not treated with IR 10 Gy, at 24 h, in DDP-sensitive and -resistant F3 and Brca1−/− cells. Rad51 foci were counted as described in Material and Methods and expressed as the mean ± SD of at least 100 nuclei in five different areas of the samples. mRNA expression levels of *REV7*, *TP53BP1* and *SHLD1* in F3 ddpR (green) (**B**) and Brca1−/− ddpR (orange) cells (**C**). Data are the mean+ SD of three independent experiments, run in triplicate and normalized on the actin gene. Data are expressed as foldchanges over the corresponding parental cell line (dotted line). (**D**) Dose–response curve of UV exposure in F3 (blue line), F3 ddpR (green line), Brca1−/− (pink line) and Brca1−/− ddpR cells (orange line). Data are the mean ± SD of three independent experiments, run in sestuplicate. (**E**) Normalized UV-damaged luciferase reporter expression after transfection in F3 (blue) and F3 ddpR (green) cells, as specified in Materials and Methods. Data are the mean ± SD of two independent experiments, run in quadruplicate. An unpaired *t*-test was used for statistical analyses. Only statistically significant differences are reported. * = *p* < 0.05; ** = *p* < 0.005; *** = *p* < 0.0005; **** = *p* < 0.0001.

**Figure 4 ijms-25-03049-f004:**
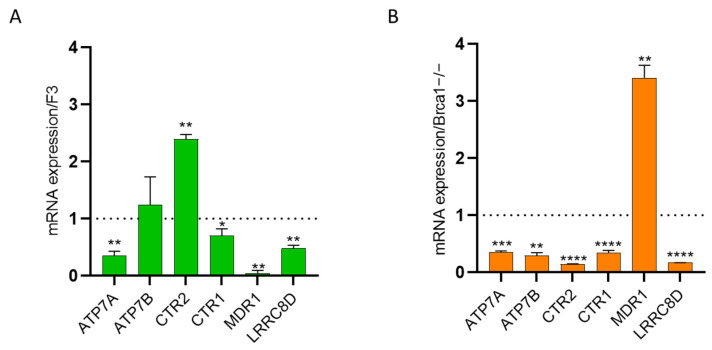
mRNA levels of *ATP7A*, *ATP7B*, *CTR2*, *CTR1*, *MDR1* and *LRRC8D* in F3 ddpR (green) (**A**) and Brca1−/− ddpR (orange) cells (**B**). Values are the mean + SD of three independent experiments, run in triplicate and normalized on the actin gene. Data are expressed as foldchange over the corresponding parental cell line (dotted line). For statistical analyses, unpaired *t*-test was used. Only statistically significant differences are reported. * = *p* < 0.05; ** = *p* < 0.005; *** = *p* < 0.0005; **** = *p* < 0.0001.

**Figure 5 ijms-25-03049-f005:**
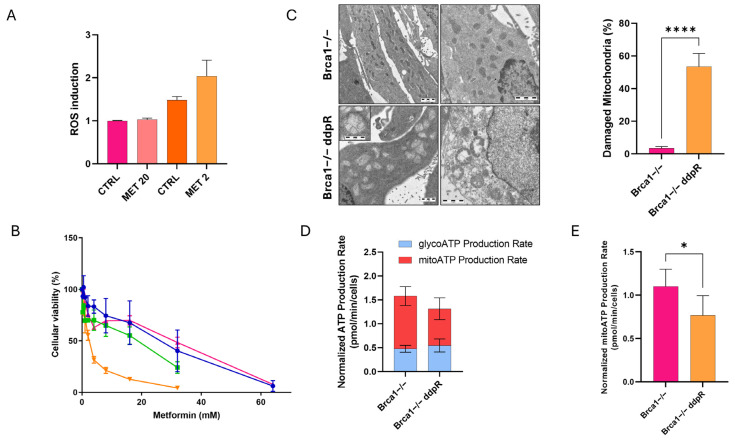
Oxidative stress in ID8 Brca1−/−-sensitive and -resistant cells. (**A**) ROS induction in Brca1−/− (pink) and Brca1−/− ddpR (orange) cells at basal levels and after 24 h IC50 dose metformin treatment. (**B**) Dose–response curve of metformin in F3 (blue line), F3 ddpR (green line), Brca1−/− (pink line) and Brca1 −/− ddpR cells (orange line). Data are the mean ± SD of three independent experiments conducted in sestuplicate. (**C**) Transmission electron microscopy of mitochondria in Brca1−/− and Brca1−/− ddpR cells (scale bar: 1 µm), and quantification of the percentage of damaged mitochondria. (**D**) Normalized ATP production rate in Brca1−/− and Brca1−/− ddpR cells. (**E**) Mitochondrial ATP production rate in Brca1−/− and Brca1−/− ddpR cells. For statistical analyses, unpaired t-test was used. * = *p* < 0.05, **** = *p* < 0.0001.

**Figure 6 ijms-25-03049-f006:**
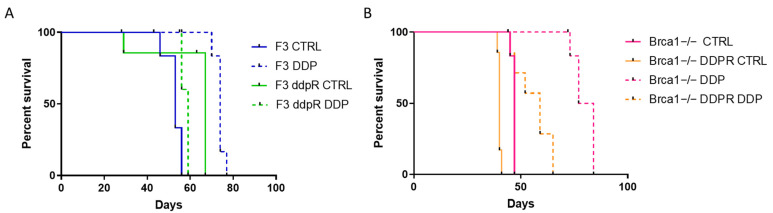
(**A**) Log-rank (Mantel–Cox) survival curve of F3- and F3 ddpR cells transplanted in mice treated with cisplatin or vehicle. (**B**) Log-rank (Mantel–Cox) survival curve of Brca1−/− and Brca1−/− ddpR tumor-bearing mice treated with cisplatin or vehicle.

**Table 1 ijms-25-03049-t001:** IC50 of different compounds in DDP-sensitive and -resistant HR-proficient and -deficient ID8 cells.

DRUG	F3	F3 ddpR	FI	*p* Value *	Brca1−/−	Brca1−/− ddpR	FI	*p* Value *
**CISPLATIN (µM)**	16.94 ± 8.26	>150	>10	***p* = 0.003**	3.88 ± 0.86	22.05 ± 11.63	5.7	***p* = 0.02**
**CARBOPLATIN (µM)**	60.67 ± 9.08	1060 ± 176.64	17.5	***p* = 0.015**	25.2 ± 0.35	86.7 ± 1.27	3.4	***p* = 0.002**
**OXALIPLATIN (µM)**	79.08 ± 17.92	212 ± 82.72	2.7	***p* = 0.05**	14.41 ± 2.18	32.55 ± 3.56	2.1	***p* = 0.0078**
**OLAPARIB (µM)**	6.44 ± 0.5	7.53 ± 3.98	1.2	*p* = 0.63	0.08 ± 0.002	2.05 ± 0.7	24.5	***p* = 0.05**
**RUCAPARIB (µM)**	13.68 ± 2.86	5.24 ± 2.63	0.4	***p* = 0.0197**	0.21 ± 0.1	3.58 ± 1.71	17.3	***p* = 0.027**
**NIRAPARIB (µM)**	0.35 ± 0.04	0.33 ± 0.3	0.9	*p* = 0.93	0.04 ± 0.01	0.26 ± 0.07	7.5	***p* = 0.043**
**YONDELIS (nM)**	0.8 ± 0.36	1.95 ± 0.74	2.5	*p* = 0.18	0.11 ± 0.02	0.57 ± 0.04	5.2	***p* = 0.0056**
**PACLITAXEL (nM)**	19.72 ± 2.31	4.18 ± 1.8	0.2	***p* = 0.017**	22.27 ± 1.9	14.8 ± 3.1	0.7	*p* = 0.1
**DOXORUBICIN (nM)**	84.62 ± 7.01	44.2 ± 0.49	0.5	***p* = 0.01**	34.8 ± 10.7	27.65 ± 0.2	0.8	*p* = 0.44
**KU55933 (µM)**	4.06 ± 0.59	2.42 ± 0.16	0.6	*p* = 0.06	1.66 ± 0.77	1.11 ± 0.18	0.7	*p* = 0.43
**AZD6738 (µM)**	0.26 ± 0.07	0.83 ± 0.65	3.2	*p* = 0.2	0.2 ± 0.04	0.23 ± 0.11	1.2	*p* = 0.64
**AZD1775 (µM)**	0.42 ± 0.2	1.08 ± 0.69	2.6	*p* = 0.11	0.2 ± 0.09	0.2 ± 0.14	0.9	*p* = 0.76
**AZD7762 (µM)**	0.19 ± 0.08	0.38 ± 0.17	2	*p* = 0.08	0.13 ± 0.06	0.22 ± 0.15	1.7	*p* = 0.28
**NU7741 (µM)**	1.72 ± 0.11	1.34 ± 0.07	0.8	*p* = 0.06	**2.32 ± 0.11**	**1.02 ± 0.29**	**0.4**	***p* = 0.03**

FI: fold increase. * Unpaired *t* test.

## Data Availability

Data is contained within the article and Appendix A.

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
