# Peer review of "Different Patterns of Platinum Resistance in Ovarian Cancer Cells with Homologous Recombination Proficient and Deficient Background"

_ijms, 2024, doi:10.3390/ijms25053049_

Round 1

Reviewer 1 Report

Comments and Suggestions for Authors

The authors present interesting evidence that resistance mechanisms in ovarian cancer adopt a variety of mechanisms including reprogramming of DDR. There are a couple of issues that should be addressed by the authors.

1) They use a host cell assay to confirm changes in NER in the F3 ddrp line, but they do not confirm protein changes that could be used for patient lines to detect this reprogramming. They mention in the discussion that ERCC1 is often used, but testing this protein expression or even XPG expression would be a way to translate the results to patients. They already note functional tests are not possible, but this linkage is critical for enhancing the significance of the findings.

2) The characterization of the HR increase and the NHEJ decrease in the ID8 cells is incomplete. The RAD51 images need to be presented in Figure 3 to evaluate the foci formation. Additionally, a host cell reactivation assay for NHEJ or probing more common NHEJ proteins like DNAPKcs or Ku would be helpful. Did the authors consider testing a DNAPKcs inhibitor? These results would enhance the Wee, ATM, and ATR results.

3) The authors should include a more thorough description of the F3 metformin results in that section. Lines 233-235 should be presented similar to lines 228-230. Additionally, the authors do not explain the mitochondrial damage they observe in the ID8 resistant cell line. They state there is slower growth, but the defect in ATP production and mitochondria is significant, so why are the mitochondria so damaged, and how is the cell compensating energetically? Increase fatty acid synthesis? Showing the impact on pro- and anti-apoptotic mitochondrial protein in that cell line would also significantly enhance the findings and mechanisms of resistance.

Reviewer 2 Report

Comments and Suggestions for Authors

The novelty of the manuscript is good. The in vitro and in vivo of experimental data is sufficient. I recommend publication of the manuscript. I suggest making a minor revision as following:

The author can explain why the ID8 F3 (Tp53−/−) are the HR proficient and why ID8 Brca1-/- (Tp53-/-; Brca1-/-),are HR deficient cell lines in the introduction part or in the method part, though the author mention it in discussion part. In that way, the reader can easily under the experimental design. 

Reviewer 3 Report

Comments and Suggestions for Authors

The Authors report findings of a study on platinum resistance in ovarian carcinoma. 

I suggest some minor changes to improve the quality of the manuscript.

Language: Some sentences are somewhat convoluted, and there is room for improvement in terms of clarity and precision in language. Ensure that each sentence conveys a clear message.

Keywords: The keywords are relevant, but consider adding terms that specifically address the methods used, such as "isogenic cell system" or "DNA repair assays."

Introduction:

While the background information is valuable, consider streamlining the introduction to focus more directly on the research question and the need for the current study.

Clearly state the objectives of the study within the introduction to guide readers on what to expect in the subsequent sections.

Results:

The results section is detailed and includes relevant data. However, some improvements can be made:

-       Consider reorganizing the results section for better flow, grouping related findings together and providing clear transitions between subsections.

-       When presenting statistical results, provide the relevant p-values and indicate the significance level. Additionally, consider using consistent terminology when reporting statistical significance (e.g., p < 0.05).

-       Some sentences could be rephrased for clarity, especially when presenting numerical data.

Discussion:

The discussion provides a comprehensive interpretation of the results, but the following suggestions may enhance its clarity and impact:

Begin the discussion by summarizing the key findings of the study before delving into detailed interpretations.

Establish clear connections between your findings and existing literature. Discuss how your results contribute to the current understanding of platinum resistance in ovarian cancer.

Clearly discuss the implications of your findings and acknowledge any limitations in the study.

References: ref.1 should be updated with 2024 publication

Comments on the Quality of English Language

Some sentences are somewhat convoluted, and there is room for improvement in terms of clarity and precision in language. Ensure that each sentence conveys a clear message.
